# Self-Management Support with Yoga on Psychological Health and Quality of Life for Breast Cancer Survivors

**DOI:** 10.3390/ijerph19074183

**Published:** 2022-03-31

**Authors:** Oksoo Kim, Narae Heo

**Affiliations:** 1College of Nursing, Ewha Womans University, Seoul 03760, Korea; ohong@ewha.ac.kr; 2Department of Nursing, Hansei University, Gunpo-si 15852, Gyeonngi-do, Korea

**Keywords:** breast cancer, program development, yoga, quality of life, self-management

## Abstract

The purpose of this study was to identify whether the improvement of self-management support (SMS) combined with yoga can improve anxiety, depression, stress, and quality of life in female breast cancer survivors. The SMS combined with yoga group and control group (yoga only) were comprised of 21 and 20 participants, respectively. The SMS with yoga and yoga programs were provided to each group for 8 weeks. Data obtained from self-report questionnaires included details of anxiety, depression, stress, and quality of life. The levels of anxiety, depression, and stress significantly decreased in both the SMS with yoga and control groups (*p* < 0.05). Moreover, the quality of life improved significantly in the SMS with yoga group (*p* < 0.001). Among the five quality-of-life domains (physical, social/family, emotional and functional well-being, and breast cancer subscale), social/family well-being in the SMS with yoga group was significantly higher than that in the yoga-only group (*p* = 0.011). Conclusions: The results show that yoga is a beneficial activity for the psychological health of breast cancer survivors. Our findings suggest that SMS strategies can improve quality of life.

## 1. Introduction

Breast cancer is the most prevalent cancer in the world, with 7.8 million women diagnosed between 2015 and 2020 [1]. In 2018, according to the Korea Central Cancer Registry, breast cancer patients ranked second after thyroid cancer among female cancers [2]. Although the incidence of breast cancer is high, the survival rate is also increasing. The 5-year survival rate for women with breast cancer in 2014–2018 was 93.3%, which is an increase of about 2.2% from 2006–2010 [2]. These data show that patients with breast cancer have a higher survival rate now than in previous years.

After medical treatment, breast cancer survivors experience physical and functional symptoms such as fatigue, pain, lymphedema, deconditioning-related functional changes, and cognitive dysfunction [3,4]. In addition, breast cancer survivors complain of psychosocial difficulties such as depression, anxiety, stress, anger, and stigma due to treatment side effects such as hair loss, body changes, comorbidity, and the possibility of recurrence or metastasis [5,6,7]. These psychological problems often begin at the diagnosis of breast cancer and are the most common late effects that breast cancer survivors experience [8]. As such, the various physical symptoms experienced during the recovery process and psychological problems such as anxiety and depression are related closely to a deterioration in the quality of life among breast cancer survivors [9,10]. These psychological problems act as negative factors that reduce the well-being of breast cancer survivors and delay adaptation and the recovery of daily life.

Following treatment, female breast cancer survivors attempt to overcome the later effects of psychological problems through physical activities such as yoga [11,12]. Researchers have proposed that yoga is a safe, optimal form of exercise for breast cancer survivors who have completed their treatment because yoga improves functional ability and psychological health [12,13]. Although physical activity such as yoga can ameliorate health problems, breast cancer survivors continuously encounter perceived barriers to regularly practicing physical activity [14]. 

For this reason, Davies and Batehup [15] proposed a self-management support (SMS) strategy to reduce barriers and maintain compliance with health behaviors among cancer survivors. Previous studies have shown the feasibility and utilization of various self-management strategies. Goal setting is an essential strategy to elicit individual expectations related to health improvement [16,17]. Molassiotis et al. [18] provided self-report logs that could help patients recognize and monitor their health status in achieving their goals. Short-message delivery services can also facilitate the provision of health information and remind patients to comply with guidelines. In particular, Husson et al. [19] reported that providing information to cancer survivors can also improve the quality of life.

In a study by Bruera et al. [20] cancer patients who used a telephone intervention for taking daily medications over 14 days, experienced decreased fatigue, nausea, depression, anxiety, and drowsiness. In addition, researchers reported cancer patients can benefit from face-to-face contact and sharing methods to cope with their subjective perceptions of their lives [21,22]. The strategies are based on Bandura’s [23] theory of self-efficacy. In other words, the self-efficacy of breast cancer survivors strengthens health behavior by integrating strategies designed to improve self-management. As a result, it is directly related to health.

Based on the benefits of yoga on psychological health and the advantages of self-management strategies, we devised SMS with yoga. This study used the following strategies based on previous studies: goal-setting, self-monitoring, provision of information, short-message delivery, positive feedback, technology-assisted delivery, sharing experience, and face-to-face contact. The procedure of this method (SMS with yoga program) will be explained in more depth below (Section 2.3).

This study aimed to identify the benefits of yoga and SMS combined with yoga on anxiety, depression, stress, and quality of life among breast cancer survivors for 8 weeks. The study hypotheses were as follows: (a) at the endpoint, participants in the SMS with yoga group will report decreased anxiety, depression, and stress compared to those in the yoga group; and (b) at the endpoint, participants in the SMS with yoga group will report enhanced quality of life compared to those in the yoga group.

## 2. Materials and Methods

### 2.1. Participation Criteria

To meet the inclusion criteria, potential participants were required to have completed chemotherapy or radiation therapy following surgery at least 4 weeks before the study. Potential participants with a history of systemic and/or musculoskeletal disorders and those who practiced yoga or moderate to vigorous physical activity within 4 weeks before the study were excluded from participation. Regardless of race, age, and socioeconomic status, participants were recruited voluntarily as those who wanted to participate in the study.

### 2.2. Data Collection

The Institutional Review Board (IRB) of Ewha Womans University Hospital approved this study protocol (No. ECT 14-01A-34). Breast cancer survivors were recruited from the cancer center for women in Korea through posters and pamphlets in the hospital.

The researcher explained to subjects the purpose and method of this study and the research process, including self-report questionnaires. Confidentiality was guaranteed in that no identifying characteristics were displayed on the self-report questionnaire form and all data were reported in aggregate form. The participation of subjects was completely voluntary and subjects were able to withdraw all data at any time. Completed questionnaire forms and all data will be kept in a locked file cabinet in the researcher’s office. In addition, the IRB contact information was provided to participants so that the rights of the participants could be protected and research could be inquired. All participants provided informed consent and the study was conducted in accordance with the Declaration of Helsinki.

First, a participant number was assigned in the order by which the informed consent to participate was received. Second, participants were divided into five groups according to the cancer stage. The participants were divided into experimental and control groups. In the order of the numbers, participants were assigned sequentially to groups as follows: Participant 1 was assigned to the experimental group, Participant 2 was assigned to the control group, Participant 3 was assigned to the experimental group, and so on.

A total of eight participants dropped out of the program. Finally, a total of 41 participants completed the program after 8 weeks.

### 2.3. Program Development

The SMS with yoga program was developed after an extensive literature search of relevant material [24]. The program was then evaluated and revised by six healthcare professionals: two medical doctors, a nursing professor, a head nurse, and two physical education experts. The devised program in this study was partitioned into two parts: SMS and yoga. The SMS combined with yoga program was provided to the experimental group for 8 weeks; the yoga-only program was provided to the control group.

#### 2.3.1. Yoga Program

The yoga developed in this program was reviewed by the six health professionals and was based on Hatha yoga. Hatha yoga is a widely recognized modern yoga practice that is characterized by maintaining body postures for a relatively long time and may include breathing and meditation [25,26]. The body postures generally consist of various poses that imitate natural shapes such as sun, moon, and trees, and every posture is taken through a sequence of slow movements to increase blood flow and flex muscles [27]. The postures used in this study were based on five anatomical poses—flexion, extension, rotation, abduction, and adduction—that represent the basic skeleton of the human body. Postures were based on basic movements, and each posture could be modified depending on the body of each patient [28]. The yoga postures were composed of pictures on posters so that yoga could be performed alone at home for 8 weeks for 50 min, three times a week. At the first visit, participants in both groups received the yoga poster and were instructed to perform the yoga program at home.

#### 2.3.2. SMS with Yoga

The SMS with yoga schema in this study supports the design of SMS guidance for developing interventions for cancer survivors by Macmillan Cancer Support, the Department of Health, and the National Health Service [15]. In this study, subjects in the experimental group engaged in regular yoga for 8 weeks using three delivery methods (information provision, sharing experience, and positive feedback) and four intervention techniques (goal-setting, self-monitoring, face-to-face contact, and technology-assisted delivery).

First, goal setting and self-monitoring strategies were applied. At the first visit, the participants were asked to write down their ultimate goal of yoga practice, and a diary was provided to ensure that yoga was practiced three times weekly.

Second, an information-provision strategy through mobile instant messaging was applied. Information on the importance of exercise and weight control (Week 2), exercise considering the physical condition (Week 4), and improving physical and mental health (Weeks 5 and 7) were provided.

Third, positive feedback and telephone-assisted support strategies were applied. The researcher checked the health of the participants by phone on Week 3 and Week 6 and encouraged them to practice yoga.

Fourth, experience sharing and face-to-face contact strategies were applied. At the 4th and 8th weeks, the participants visited the hospital to share their experiences related to yoga practice.

### 2.4. Outcome Measures

All measures have been verified by Koreans and have been used to measure the psychology of Koreans. Researchers directly provided the self-report questionnaires to the participants before and after the start of the questionnaires program at Ewha Womans University Hospital. The self-report questionnaires were used to measure participant anxiety, depression, stress, and quality of life. Participants were fluent in Korean, understood the questionnaire, and responded to the questionnaire themselves. Participants completed the questionnaires in approximately 15 to 20 min.

#### 2.4.1. Anxiety

Anxiety was measured using the Korean version [29] of the State-Trait Anxiety Inventory (STAI) [30], which includes measurement items for the state and trait anxiety domains. In this study, the state anxiety domain was applied to measure the anxiety an individual felt at the time of the study. The state anxiety domain contains 20 items measured with a 4-point Likert-type scale, and all items are rated on a scale of 1 point for ‘not at all’ to 4 points for ‘very much so’, with a total score of 20 to 80 points. A higher total score indicates greater anxiety. Cronbach’s alpha in this study was 0.85.

#### 2.4.2. Depression

Depression was measured using the Korean version [31] of the Beck Depression Inventory [32], which includes emotional, psychological, and physical symptoms of depression and assesses the presence and severity of symptoms. The scale consists of 21 items measured with a 4-point Likert-type scale and each item is scored on a scale of 0 to 3 points. Overall scores ranged from 0 to 63 with a higher score indicating more severe depression. Cronbach’s alpha in this study was 0.81.

#### 2.4.3. Stress

Stress was measured using the Korean version [33] of the Perceived Stress Scale [34], which measures the degree to which one perceives life as stressful. The scale includes 14 items measured with a 5-point Likert-type scale and all items are rated on a scale of 0 points for ‘never’ to 4 points for ‘very often’, with a total score ranging from 0 to 56. A higher score indicates more severe stress. In this study, Cronbach’s alpha was 0.85.

#### 2.4.4. Quality of Life

Quality of life was measured using the Korean version of the Functional Assessment of Cancer Therapy-Breast Cancer Version 4 (FACT-B), which was developed by the Functional Assessment of Chronic Illness Therapy [35] for measuring health-related quality of life. The questionnaire is divided into five domains: physical well-being (7 items), social/family well-being (7 items), emotional well-being (6 items), functional well-being (7 items), and breast-cancer subscale (10 items). The measurement is comprised of 37 questions measured with a 5-point Likert-type scale and each domain is scored from 0 points for ‘not at all’ to 4 points for ‘very much’. An overall score can range from 0 to 148 points with a higher score indicating higher quality of life. In this study, Cronbach’s alpha was 0.91.

### 2.5. Statistical Anlayses

SPSS version 22.0 (IBM) was used for data analyses. Baseline characteristics for homogeneity were compared between the experimental and control groups using independent *t*-tests and Fisher’s exact tests. Paired *t*-tests were used for within-group measures at pre- and post-tests. Independent *t*-tests were used to test for group differences regarding psychological health (anxiety, depression, and stress) and quality of life due to the intervention. A statistical power analysis was performed to determine power for this study. The G*power 3.0 software was used to calculate the sample size [36]. The required sample size was calculated based on a significance of 0.05, and a power of 0.8. In total, 41 participants completed the program.

## 3. Results

### 3.1. Participant Characteristics at Baseline

Table 1 shows the participant characteristics (age, age group, marital status, education, type of household, job, income, religion, time of diagnosis, stage of cancer, type of surgery, and type of treatment received) at baseline. No significant differences were found between groups across all variables (*p* > 0.05) at baseline.

### 3.2. Participant Anxiety, Depression, and Stress Levels

The anxiety, depression, and stress levels of the experimental group participants significantly decreased *(t* = 5.72, *p* = 0.001; *t* = 7.05, *p* = 0.001; *t* = 5.65, *p* = 0.001, respectively) after the 8-week intervention (Table 2). Significant differences (*t* = 2.68, *p* = 0.015; *t* = 2.55, *p* = 0.02; *t* = 2.20, *p* = 0.041, respectively) were also found in the three domains (i.e., anxiety, depression, and stress) for the control group at the pre- and post-test periods. However, no significant differences (*p* > 0.05) in all three domains (i.e., anxiety, depression, stress) were found between the experimental group (SMS with yoga group) and control group (yoga-only group).

### 3.3. Participant Quality of Life

The statistical differences of the quality of life were shown in Table 3. There was a statistically significant difference in the total quality of life in the SMS with yoga group (*t* = −5.86, *p* < 0.001). In the five quality-of-life domains, including physical well-being, social/family well-being, emotional well-being, functional well-being and breast cancer subscale, significant differences in the SMS-with yoga group were found in all subscales from pre- to posttest at 8 weeks (*t* = −2.37, *p* = 0.028, *t* = −4.90, *p* < 0.001; *t* = −3.89, *p* = 0.001; *t* = −3.67, *p* = 0.002; *t* = −4.57, *p* < 0.001; Table 3). In addition, significant differences in physical and functional well-being subscales were also found in the in yoga-only group over the 8 weeks (*t* = −2.26, *p* = 0.036; *t* = −2.28, *p* = 0.034; Table 3). Among the quality-of-life subscales, social/family well-being in the SMS with yoga group was significantly higher than that in the yoga-only group (*t* = 2.68, *p* = 0.011; Table 3).

## 4. Discussion

The first hypothesis of this study was that participants in the SMS with yoga group would report decreased anxiety, depression, and stress compared to the yoga group at the endpoint. The first hypothesis was not supported. In this study, the difference between the groups was not statistically significant. No other research has operated a yoga program by applying various SMS strategies as in this study, so an accurate comparison of the results is not possible. These results are attributed to the lack of special SMS strategies for the psychological health promotion of breast cancer survivors in this study. In particular, breast cancer survivors complained of psychological difficulties due to concerns such as loss of physical image, difficulty returning to work, a decline in economic income, and changes in social and family status. Psychological and healthcare support needs persist from the time of breast cancer diagnosis to long-term survival of more than 5 years [37]; therefore, healthcare providers must identify the psychological needs of patients to provide specialized SMS strategies.

Despite the lack of statistical significance between groups, through the 8-week program, the results showed statistical differences for anxiety, depression, and stress in both groups. The results of this study are consistent with the results of the study by Olsson Möller et al. [38] that focused on reducing depression and anxiety through interventions with yoga after breast cancer treatment. Similarly, an analysis of 38 articles by Galliford et al. [39] supported the assertion that yoga therapy reduces stress and improves psychological health. In light of the previous findings of other studies, the results in this study showed that the yoga program had beneficial effects on the psychological health of breast cancer survivors; therefore, yoga can play a latent role in relieving the psychological distress experienced by breast cancer survivors (i.e., depression and anxiety), which can impact psychological health. Above all, yoga is a physical activity that can be easily performed at home regardless of time and place, and research results have reported incessantly that yoga is effective in coping with anxiety, depression, and stress [40,41,42]. Therefore, providing a service that allows breast cancer survivors to learn yoga alone can improve their psychological health. Above all, this strategy could contribute to the improvement of medical service efficiency and the convenience of physical activities.

The second hypothesis in this study stated that participants in the SMS with yoga group would report enhanced quality of life compared to the yoga group. The second hypothesis was supported only in the social/family well-being domain of the quality of life. Among quality-of-life domains, social/family well-being in the SMS with yoga group was significantly higher than in the yoga-only group. These findings were likely to be mirrored by the benefits of SMS application, considering that participants in the control group had no SMS. The results show that self-management strategies promote social/family well-being and are important healthcare services for breast cancer survivors. In particular, this could be applied as a way to promote social/family well-being in the physical activity process. Considering the results of previous studies on the correlation between social support, quality of life, and physical activity [43], generalizing the results of this study by increasing the number of samples in a similar study in the future is necessary.

However, there were no differences between groups in other subdomains except for social/family well-being. These results may reflect the daily life of breast cancer survivors. Survivors of breast cancer experience various difficulties. These difficulties include concerns about the possibility of disease progression after discharge, household income, child-care, bathing, difficulty wearing clothes, and physical changes [44,45,46]. Therefore, it is necessary to provide customized strategies individually by identifying various unmet supportive management needs of breast cancer survivors.

Our results show that quality of life scores for all five domains (physical well-being, social/family well-being, emotional well-being, functional well-being, and breast cancer subscale) were significantly improved in the SMS with yoga group (114.90 ± 14.91) and yoga group (107.70 ± 23.57) after the intervention. From the start to the end of this study, the experimental group’s quality-of-life score was more than eight points higher than the quality-of-life score reported in the study of breast cancer survivors by Kwon and Yi [47]. A possible reason for the difference in scores of these two studies may be that recruitment was conducted in separate locations. In this study, participants attended a hospital visit and participated in an SMS program, whereas participants in the study by Kwon and Yi took part in a community-setting self-support group. One can infer that the scores are better for participants in the group who, albeit not often, received regular checkups, and follow-up supported management in contrast to the support group in a community setting. In particular, this shows that regular follow-up strategies are needed in specialized institutions to maintain or increase patient quality of life. On the other hand, the SMS with yoga group and the yoga-only group showed significant results in two quality-of-life domains (physical and functional well-being) at 8 weeks, although there was no difference between the two groups. Yoga is an activity that increases physical function and reduces physical restrictions [48]; thus, yoga may have had a positive impact on improving the quality of life. Accordingly, it is expected that the research and development of smart human yoga services tailored to the body will be necessary with the industrialization of ICT-based smart healthcare services reflecting social changes.

Research on SMS resources for cancer survivors has been conducted actively for the last 10 years, with indications that multidimensional support strategies have a positive influence on the health of cancer survivors [49]. In addition, the results of this study support the results of these previous studies. This suggests that SMS plays an important role in reducing anxiety, depression, and stress and improving the overall quality of life in various situations following breast cancer. Therefore, various support strategies to improve the psychological health and quality of life of breast cancer survivors must be developed continuously. However, there is a limitation in that the result has not been able to verify accurately whether self-support strategies have improved psychological health and the quality of life by improving the ability of self-management. In the future, studies on the mediating and moderating effects are needed, and the results will help expand related theories applicable to practice.

## 5. Conclusions

Our findings show that (a) yoga improves psychological health and (b) SMS combined with yoga improves the social well-being of quality of life. Provision of care by designing a variety of accessible and usable strategies to improve the psychological health and quality of life of breast cancer survivors is important. In particular, the various SMS strategies applied in this study provide a direction for the customized health management of breast cancer survivors. Healthcare services that meet various levels of patient needs are essential, and creative strategies for providing convergent services that align with the diverse needs of the patient are needed. These strategies will improve the self-management ability of the patient, eventually ensuring their health and quality of life.

## Figures and Tables

**Table 1 ijerph-19-04183-t001:** Participants’ characteristics at baseline.

Characteristics	Categories	Exp. ^1^ (n = 21)	Cont. ^2^ (n = 20)	*t* or F	*p*
		n (%)			
Age (years)	M ± SD	46.62 ± 7.09	45.75 ± 6.60	0.41	0.687
Age group (years)	<40	3 (14.3)	1 (05.0)	1.02	0.866
	40–49	10 (47.6)	12 (60.0)		
	50–59	7 (33.3)	5 (25.0)		
	≥60	1 (4.8)	2 (10.0)		
Marital status	Married	19 (90.5)	18 (90.0)	0.00	0.959
	Unmarried	2 (09.5)	2 (10.0)		
Education	High school or less	7 (33.3)	8 (40.0)	0.20	0.658
	Completed university	14 (66.7)	12 (60.0)		
Type of household	2 or more persons	21 (100.0)	19 (95.0)	1.08	0.300
	Living alone		1 (5.0)		
Job	Yes	9 (42.9)	9 (45.0)	0.02	0.890
	No	12 (57.1)	11 (55.0)		
Income	Less than 2000	3 (14.3)	2 (10.0)	0.18	0.916
(USD/month)	2000–5000	12 (57.1)	12 (60.0)		
	5000 or more	6 (28.6)	6 (30.0)		
Religion	Yes	9 (42.9)	12 (60.0)	1.21	0.272
	No	12 (57.1)	8 (40.0)		
Time since diagnosis (months)	M ± SD	16.19 ± 12.90	16.10 ± 0.20	0.03	0.982
Stage of cancer	0	1 (04.8)	1 (05.0)	1.39	0.922
	I	8 (38.1)	5 (25.0)		
	II	7 (33.3)	8 (40.0)		
	III	4 (19.0)	5 (25.0)		
	IV	1 (04.8)	1 (05.0)		
Type of surgery	Breast-conserving surgery	19 (90.5)	17 (85.0)	0.29	0.592
	Mastectomy	5 (09.5)	3 (15.0)		
Type of treatment received	Radiation therapy	5 (23.4)	4 (20.0)	0.09	0.534
	Chemotherapy with radiation	16 (76.2)	16 (80.0)		

^1^ Exp.—experimental group, ^2^ Cont.—control group.

**Table 2 ijerph-19-04183-t002:** Participant anxiety, depression, and stress level.

	Pretest (First Visit)	Posttest (Last Visit)	Posttest-Pretest (Within-Group)	Exp.-Cont. ^3^ (Between-Groups Comparison)
	M ± SD	M ± SD	*t* (*p*)	*t* (*p*)
Anxiety				
Exp. (n = 21)	46.23 ± 10.31	36.90 ± 7.91	5.72 (0.001)	1.34 (0.188)
Cont. (n = 20)	44.50 ± 08.50	38.75 ± 9.36	2.68 (0.015)	
Depression				
Exp. (n = 21)	12.61 ± 06.46	7.14 ± 5.31	7.05 (0.001)	0.86 (0.398)
Cont. (n = 20)	12.60 ± 06.21	8.60 ± 4.69	2.55 (0.020)	
Stress				
Exp. (n = 21)	24.52 ± 07.94	15.28 ± 5.05	5.65 (0.001)	1.58 (0.122)
Cont. (n = 20)	25.50 ± 07.63	20.60 ± 8.67	2.20 (0.041)	

^3^ Exp.-Cont.—Difference between the experimental group and the control group.

**Table 3 ijerph-19-04183-t003:** Participant quality of life.

	Pretest (the First Visit)	Posttest (the Last Visit)	Posttest-Pretest (Within-Group)	Exp.-Cont. ^4^ (Between-Groups Comparison)
	M ± SD	M ± SD	*t* (*p*)	*t* (*p*)
Quality of life				
Total score				
Exp. (n = 21)	96.00 ± 17.68	114.90 ± 14.91	−5.86(<0.001)	1.09(0.282)
Cont. (n = 20)	96.26 ± 20.90	107.70 ± 23.57	−1.875(0.077)	
Physical well-being				
Exp. (n = 21)	21.43 ± 5.28	23.76 ± 4.76	−2.37 (0.028)	0.68 (0.499)
Cont. (n = 20)	20.65 ± 6.20	24.25 ± 3.07	−2.26 (0.036)	
Social/family well-being				
Exp. (n = 21)	16.71 ± 5.28	22.43 ± 3.45	−4.90 (<0.001)	2.68 (0.011)
Cont. (n = 20)	16.65 ± 5.00	17.70 ± 7.16	−0.81 (0.427)	
Emotional well-being				
Exp. (n = 21)	17.67 ± 3.97	20.57 ± 3.26	−3.89 (0.001)	1.15 (0.256)
Cont. (n = 20)	17.10 ± 4.31	18.30 ± 4.33	−0.92 (0.367)	
Functional well-being				
Exp. (n = 21)	16.48 ± 6.98	20.76 ± 5.79	−3.67 (0.002)	0.06 (0.950)
Cont. (n = 20)	17.05 ± 6.33	21.20 ± 6.23	−2.28 (0.034)	
Breast cancer subscale				
Exp. (n = 21)	23.71 ± 5.34	27.38 ± 5.01	−4.57 (<0.001)	0.71 (0.484)
Cont. (n = 20)	24.80 ± 5.86	27.10 ± 8.11	−1.28 (0.215)	

^4^ Exp.-Cont.—Difference between the experimental group and the control group.

## Data Availability

Supporting data and results can be found in the tables.

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
