# Peer review of "Self-Management Support with Yoga on Psychological Health and Quality of Life for Breast Cancer Survivors"

_ijerph, 2022, doi:10.3390/ijerph19074183_

Round 1

Reviewer 1 Report

To the authors- I commend you on a generally well written manuscript. There are several suggested edits in an attached word document for your review. I think your work will contribute to a growing body of well designed research in breast cancers survivors and it will ultimately improve health and health outcomes. Kind regards and best of luck with your future work. 

Author Response

March 17, 2022
Reviewer of International Journal of Environmental Research and Public Health

Dear Reviewer:

Above all, thank you for giving us the opportunity to revise our article.
We edited the article fully reflecting the comments. 

We appreciate the reviewers’ comments. We believe that the quality of our article has improved based on the comments.

The COVID-19 pandemic continues to exacerbate existing problems affecting a host of areas in the medical and healthcare sector, such as limiting the reach of healthcare services and creating medical voids for the socially disadvantaged and unequal distribution of medical resources.  These problems are causing mental and social issues and inhibiting the improvement of psychological health and quality of life, including one's well-being. Against this background, it is important to tailor healthcare services to patients by designing a variety of accessible and usable strategies aimed at improving breast cancer survivors' psychological health and quality of life.

We believe that the results of the study will improve health care workers' awareness of program development that applies various management methods to breast cancer survivors, and they will seek SMS strategies for breast cancer survivors. In addition, we further believe that SMS strategies will contribute to presenting customized health care directions for breast cancer survivors.

With best wishes,
Narae Heo, MS, FNP, PhD
[email protected]

Reviewer 2 Report

see attached file

Author Response

Dear Reviewer:

Above all, thank you for giving us the opportunity to revise our article. We edited the article fully reflecting the comments

We appreciate the reviewers’ comments. We believe that the quality of our article has improved based on the comments

The COVID-19 pandemic continues to exacerbate existing problems affecting a host of areas in the medical and healthcare sector, such as limiting the reach of healthcare services and creating medical voids for the socially disadvantaged and unequal distribution of medical resources.  These problems are causing mental and social issues and inhibiting the improvement of psychological health and quality of life, including one's well-being. Against this background, it is important to tailor healthcare services to patients by designing a variety of accessible and usable strategies aimed at improving breast cancer survivors' psychological health and quality of life.

We believe that the results of the study will improve health care workers' awareness of program development that applies various management methods to breast cancer survivors, and they will seek SMS strategies for breast cancer survivors. In addition, we further believe that SMS strategies will contribute to presenting customized health care directions for breast cancer survivors.

Below are the answers to the comments.

Point 1:

Overall, authors should more deeply explain what their study adds to exiting literature. In addition, the Introduction could be more specifically pointing towards the aims. I think this is especially important given that the audience of IJERPH are not necessarily know a lot about yoga, about SMS, about SMS strategies applied to yoga. I suggest being explicit about the importance and context of your aims.

Response 1:

Thank you for your comment.

We revised to more deeply explain what our study adds to existing literature. By adding our explanation, the importance and context of our aims were explicitly expressed.

  • (p 1/12, line 30-31) These data show the breast cancer survivors have a higher survival rate now than in previous years.
  • (p 1/12, line 41-43) These psychological problems act as negative factors that reduce the well-being of breast cancer survivors and delay adaptation and recovery of daily life.
  • (p 2/12, line 67-75) Above all, the survival rate of breast cancer patients is high. Survivor’s health can be maintained through continuous management through physical activity such as yoga during the survival period. Therefore, various support strategies should be sought so that patients can continuously practice healthy behavior in their daily lives. In addition, various programs should be prepared to create synergy in health improvement by converging resources; such programs will eventually affect psychological health and quality of life. Therefore, health professionals should provide diverse services to help survivors live healthier lives and play a more active role in their health care through multimodal strategies that allow survivors to manage themselves.

Point 2:

At the end of the Introduction the authors should briefly articulate the hypotheses under investigation and their correspondence to research design, describing how these were derived from theory or are logically connected to previous data and argumentation. Clearly develop the rationale for each hypothesis on the basis of a succinct review of the literature (on SMS strategy applied to yoga, if exiting) and possibly offer a presentation of the results that were expected.

Response 2:

Thank you for your helpful comments.

A succinct review of the literature was added to show what theories were related to the research design, and the connection to the hypothesis derived was shown. At the end of the Introduction, we briefly explained the hypothesis and the aims of the study.

  • (p 2/12, line 53-75) Previous studies have shown the feasibility and utilization of various strategies. Goal setting is an essential strategy to elicit individual expectations related to health improvement [16.17]. Molassiotis et al.[18] provided self-report logs that could help patients recognize and monitor their health status to their goals. Short-message delivery services can also facilitate health information provision and remind patients to comply with guidelines. In particular, Husson, Moles et al. [19] reported that providing information to cancer survivors also improve quality of life. In a study by Bruera et al., [20] cancer patients who applied a telephone intervention experienced for taking daily medications during 14 days, decreased fatigue, nausea, depression, and anxiety, and drowsiness. In addition, researchers reported cancer patients can benefit from face-to-face contact and sharing methods to cope with their subjective perceptions of their lives[21,22]. The strategies are based on Bandura's [23] theory of self-efficacy. In other words, the self-efficacy of breast cancer survivors strengthens health behavior by integrating strategies designed to improve self-management. As a result, it is directly related to health.
  • Above all, the survival rate of breast cancer patients is high. Survivor’s health can be maintained through continuous management through physical activity such as yoga during the survival period. Therefore, various support strategies should be sought so that patients can continuously practice healthy behavior in their daily lives. In addition, various programs should be prepared to create synergy in health improvement by converging resources; such programs will eventually affect psychological health and quality of life. Therefore, health professionals should provide diverse services to help survivors live healthier lives and play a more active role in their health care through multimodal strategies that allow survivors to manage themselves
  • (p 2/12, line 53-81) For this reason, we devised SMS with yoga for breast cancer survivors based on the benefits of yoga on psychological health and the advantages of self-management strategies. In this study, the following strategies were fused based on previous studies: goal setting, self-monitoring, information provision, short-message delivery, positive feedback, technology-assisted delivery, sharing experience, and face-to-face contact. In this study, the following strategies were fused based on previous studies.
  • (p 2/12, line 83-88) The aim of this study was to identify the benefits of yoga and SMS with yoga on anxiety, depression, stress, and quality of life among breast cancer survivors for eight weeks. The study hypotheses were as follows: (a) at the endpoint, participants in SMS with yoga group will report decreased anxiety, depression, stress compared to those yoga group; and (b) at the endpoint, participants in SMS with yoga group will report enhanced quality of life compared to those yoga group.
  • The aim of this study was to identify the benefits of yoga and SMS with yoga on anxiety, depression, stress, and quality of life among breast cancer survivors for eight weeks. The study hypotheses were as follows: (a) at the endpoint, participants in SMS with yoga group will report decreased anxiety, depression, stress compared to those yoga group; and (b) at the endpoint, participants in SMS with yoga group will report enhanced quality of life compared to those yoga group.

Point 3:

(lines 31-32) “Notably, breast cancer survivors across racial and ethnic groups experience psychological problems such as anxiety, depression, and stress even after adjunctive treatment ends”. Given that the contents of this sentence are explained in the following sentences, it seems superfluous.

Response 3:

Thank you for your comment.

We deleted the sentences

Point 4:

(lines 58-59) “Therefore, various SMS strategies were applied to yoga”, this sentence is very vague and unclear. Please, add a brief description of SMS-with-yoga program, along with previous empirical studies

evidence supporting this method, this is needed to provide background of the aims. Alternatively, add in the Introduction, also in brackets, that the procedure of this method (SMS-with-yoga program) will be more deeply explain below (section 2.3)

Resonse 4:

In the introduction, we added a brief description of the SMS-with-yoga program along with previous empirical studies. We added the sentences as your comment suggested.

  • (p 2/12, line 53-63) Previous studies have shown the feasibility and utilization of various strategies. Goal setting is an essential strategy to elicit individual expectations related to health improvement [16.17]. Molassiotis et al.[18] provided self-report logs that could help patients recognize and monitor their health status to their goals. Short-message delivery services can also facilitate health information provision and remind patients to comply with guidelines. In particular, Husson, Moles et al. [19] reported that providing information to cancer survivors also improve quality of life. In a study by Bruera et al., [20] cancer patients who applied a telephone intervention experienced for taking daily medications during 14 days, decreased fatigue, nausea, depression, and anxiety, and drowsiness. In addition, researchers reported cancer patients can benefit from face-to-face contact and sharing methods to cope with their subjective perceptions of their lives[21,22].
  • (p 2/12, line 77-82) In this study, the following strategies were fused based on previous studies : goal setting, self-monitoring, information provision, short-message delivery, positive feedback, technology-assisted delivery, sharing experience, and face-to-face contact. In this study, the following strategies were fused based on previous studies. The procedure of this method (SMS-with-yoga program) will be more deeply explain below (section 2.3).

Point 5:

(lines 68-70) “sequentially assigned into either the experimental group (SMS-with-yoga program) or control group (yoga only) based on their age, cancer stage, and time since diagnosis”. Please, provide a clearer explanation of the assignment method to the experimental/control group. What were the role of age, cancer stage, time since diagnosis, order of visit?

Response 5:

Thank you for your comment.

We deleted ambiguous and unclear sentences. The sentences (line 68-70) was edited as follows. The role of the visit order was to minimize the selection bias of participants who applied together.

  • (p 3/12, line 111-117) First, the participant number was assigned in the order in which the informed consent to participate was received. Second, participants were divided into five groups according to the cancer stage. There group were divided into experimental groups and control groups. In the order of numbers, participants were sequentially assigned to groups as fol-lows: Participant 1 was assigned to the experimental group once, Participant 2 was as-signed to the control group, Participant 3 was assigned to the experimental group, and so on.

Point 6:

The authors should provide a detailed description of the major demographic characteristics of the sample (age, socioeconomic status, race/ethnicity) and how participants were selected. What about non-participants?

Response 6:

Participants voluntarily participated in the study. Regardless of race, age, and socio-economic status, participants were recruited voluntarily as those who wanted to participate in the study.This study did not limit the age, race, and socioeconomic status of the participants. In addition, all of the participants were Korean (Asian). The number of people who did not participate in the study was described in the middle of the study. We edited the sentencse as follows.

  • (p 3/12, line 95-97) Regardless of race, age, and socio-economic status, participants were recruited voluntarily as those who wanted to participate in the study.
  • (p 3.12, line 117-119) A total of eight participants dropped out of the program. Finally, a total of 41 participants completed the program for 8 weeks.

Point 7:

(lines 81-82) provide appropriate citations

Response 7:

We provided appropriate citations

  • (p 3/12, line 122-123) The SMS-with-yoga program was developed after an extensive literature search of relevant material[15-22].

Point 8:

Authors should also indicate if the version of measures adopted (STAI, BDI etc.) were validated for Korean people and related evidence supporting their psychometric properties.

Response 8:

The measures were translated into the Korean version and that is still used for Korean diagnosis and research on the same basis. There are no specific criteria that apply only to Koreans in these three areas. We edited the sentences.

  • (p 4/12, line 176-178) Anxiety was measured using Korean version [29] of the State-Trait Anxiety Inventory (STAI) [30], which includes measurement items for the state and trait anxiety domains.
  • (p 4/12, line 185-188) Depression was measured using a Korean version [29] of the Beck Depression Inventory [30], which includes emotional, psychological, and physical symptoms of depression and assesses the presence and severity of symptoms.
  • (p 4/12, line 193-194) Stress was measured using Korean version [31] of the Perceived Stress Scale [32], which measures the degree to which one perceives life as stressful.
  • (p 5/12, line 200-204) Quality of life was measured using the Korean version of the Functional Assessment of Cancer Therapy-Breast Cancer Version 4 (FACT-B), which was developed by the Functional Assessment of Chronic Illness Therapy [33] for measuring health-related quality of life (HRQOL).
  • (p 4/12, line 166-167) All measures have been verified by Koreans and have been used to measure the psychology of Koreans. There are no psychological characteristics unique to Koreans.

Point 9:

How the measures were administered? (Specify how the instrument was introduced to participants, including any instructions about completing the survey questionnaire) and to whom, how the consent form was gathered, who conducted the procedures and where the procedures were conducted.

Response 9:

We added sentences related to survey collection.

(p 4/ line 168-173) Researchers directly provided the self report questionnaires to the participants before and after the start of the questionnaires program at Ewha Womans University Hospital. The self report questionnaires were used to measure the participants’ levels of anxiety, depression, stress, and quality of life. Participants were fluent in Korean, understood the questionnaire, and reponded to the questionnaire themselves. Participants completed the questionnaires in approximately 15 to 20 minutes.

Point 10:

Line 175, add all p>.05

Response 10:

We added the “p > .05.”

  • (p 5/12, line 225) No significant differences across all variables ( p> .05) were found between groups at baseline.

Point 11:

(lines 178-184). The study indicated that anxiety, depression and stress level significantly decreased after the eight weeks intervention in both groups. Only quality of life scores resulted increased in the experimental group but not in controls. However, it is worth noting that Physical and function well-being increased in both groups.

The way how the results were described not clearly indicates the lack of differences between groups. Please, a little rewording of the results presentation is needed to more clearly describe the lack of difference between groups.

Please, generally avoid terminology related to causal relationships throughout the paper (e.g., “Effects”)

Response 11:

Words(Between-Groups Comparison) were inserted and Table 2 was separated to clearly indicate differences between groups.

We changed the word “effect” that shows the direct result of the cause. We also deleted the words in the title.

  • (p 6/12, line 236) Table 2. Participants’ anxiety, depression, and stress,

Exp. – Con. (Between-Groups Comparison)

  • (p 7/12, line 249) Table 3. Participants’ Quality of life

Exp. – Con. (Between-Groups Comparison)

  • (p 1/12, line 2) Self-Management Support with Yoga on Psychological Health and Quality of Life for Breast Cancer Survivors

Point 12:

The comments in the discussion section are not well organized

For example (lines 198-217) all these comments are not in line with the aims of the study.

The comments need to be rewritten in a more concise way to point out how the results confirm or do not confirm the original hypotheses of the study. I would suggest opening the discussion section with a clear statement about the questions raised in the Introduction, of the support or non-support for your original hypotheses.

After presenting the results, the authors should be able to evaluate and interpret their implications, especially with respect to their original hypotheses When your hypotheses were not supported, you should offer reasonable explanations.

After examining and drawing inferences from the results, you should better emphasize any theoretical or practical consequences of the results

Response 12:

We rewrote and organized all the sentences in the discussion. In the first sentence of the paragraph, we described whether the hypothesis was supported or not.

In addition, the main research results were rearranged at the top of the paragraph. We confirmed the hypothesis and described its reasoning, interpretation, and meaning. We also emphasized practical consequences of the results.

  • (p 7-8/12, line 252-280) The first hypothesis of this study was that participants in SMS with yoga group would report decreased anxiety, depression, and stress compared to the yoga group at the endpoint. The first hypothesis was not supported. In this study, the difference between the groups was not statistically significant. As in this study, there is no research that has operated a yoga program by applying various SMS strategies, so accurate comparison of the results is impossible. These results are attributed to the lack of special SMS strategies for psychological health promotion of breast cancer survivors in this study. In particular, breast cancer survivors complained of psychological difficulties due to concerns such as loss of physical image, difficulty returning to work, economic income decline, and changes in social and family status. Psychological needs and health care support needs persist from the time breast cancer diagnosis to a long-term survival of more than 5 years [37]; therefore, healthcare providers must identify patients’ psychological needs to provide specialized SMS strategies.

Despite the lack of statistical significance between groups, through the eight-week program, the results showed statistical differences for anxiety, depression, and stress in both groups. The results of this study are consistent with the results of Olsson Möller et al.’s study [38] that focused on reducing depression and anxiety through yoga interventions after breast cancer treatment. Similarly, Galliford et al.’s [39] analysis of 38 articles supported the assertion that yoga therapy reduces stress and improves psychological health. In light of the previous studies’ findings, the results in this study showed that the yoga program had beneficial effects on breast cancer survivors’ psychological health; therefore, yoga can play a latent role in relieving breast cancer survivors’ experiences of psychological distress (i.e., depression and anxiety), which impact psychological health. Above all, yoga is a physical activity that can be easily performed at home regardless of time and place; and research results have been continuously reported that yoga is effective in coping with anxiety, depression, and stress [40-42]. Therefore, providing breast cancer survivors with a service that allows them to easily learn yoga alone improve the psychological health of many breast cancer survivors. Above all, this strategy could contribute to improving medical service efficiency and convenience for patients' physical activities.

  • (p 8/12, line 303-321) The second hypothesis of this study stated that participants in SMS with yoga group would report enhanced qualify of life compared to the yoga group. The second hypothesis was supported only in the social/family well-being of the quality of life. Among quality of life domains, social/family well-being in the SMS-with-yoga group was significantly higher than that of the yoga-only group. These findings were likely mirrored by benefits of SMS application, considering that participants in the control group had no SMS. The results show that the self-management strategies promote social/family well-being and are important health care services for breast cancer survivors. In particular, it could be applied as a way to promote social/family well-being in the physical activity process. Considering the results of previous studies on the correlation between social support, quality of life, and physical activity [46], generalizing the results of this study by increasing the number of samples in a future similar study is necessary.
  • However, there were no differences between groups in other sub-domains except for social/family well-being. This results may reflect the daily life of breast cancer survivors. Breast cancer survivors experience various difficulties. These difficulties include concerns about the possibility of disease progression after discharge, household income, child-care, bathing, difficulty wearing clothes, and physical changes [47-49]. Therefore, it is necessary to individually provide customized strategies by identifying various unmet supportive management needs of breast cancer survivors.

Point 13:

What is the theoretical, clinical, or practical significance of these findings? What problems remain unresolved or arise anew because of these results?

Response 2:

We added on the limitations of our study.

  • (p 10/12, line 350-354) However, there is a limitation in that the result has not been able to accurately verify whether self-support strategies have improved psychological health and quality of life by improving self-management ability. In the future, research on mediating effects and moderating effects is needed, and the results will help expand related theories applicable to practice.
  • (p 10/12, line 357-358) Provision of care by designing a variety of accessible and usable strategies to improve breast cancer survivors’ psychological health and quality of life is important.
  • (p 10/12, line 363-366) It is time to develop creative strategies to provide convergent services that align with diverse in line with patient needs and the large trend of social change. These strategies will improve the patients’ self-management ability, eventually ensuring the patients’ health and quality of life.

Round 2

Reviewer 2 Report

Lines 53-54 Previous studies have shown the feasibility and utilization of various strategies can be reworded in “Previous studies have shown the feasibility and utilization of various self-management strategies".

Lines 67-75 to be deleted (to avoid redundancies)

Line 75- 77 “For this reason, we devised SMS with yoga for breast cancer survivors based on the benefits of yoga on psychological health and the advantages of self-management strategies" can be reworded as…

“Based on the benefits of yoga on psychological health and the advantages of self-management strategies, we devised SMS with yoga ….”

167-167 to be deleted

281-302 to be deleted or shortened

Author Response

Response to Reviewer 2 Comments

We appreciate the comments from the reviewer. Thanks to your consideration, the quality of our article has improved.

Point 1:

Lines 53-54 Previous studies have shown the feasibility and utilization of various strategies can be reworded in “Previous studies have shown the feasibility and utilization of various self-management strategies".

Response 1:

We edited the sentence.

  • (p 2/12, line 53-54) Previous studies have shown the feasibility and utilization of various self-management strategies.

Point 2:

Lines 67-75 to be deleted (to avoid redundancies)

Response 2:

We deleted the sentences

Point 3:

Line 75- 77 “For this reason, we devised SMS with yoga for breast cancer survivors based on the benefits of yoga on psychological health and the advantages of self-management strategies" can be reworded as…

“Based on the benefits of yoga on psychological health and the advantages of self-management strategies, we devised SMS with yoga ….”

Response 3:

We edited the sentence.

  • (p 2/12, line 68-69) Based on the benefits of yoga on psychological health and the advantages of self-management strategies, we devised SMS with yoga.

Point 4:

167-167 to be deleted

281-302 to be deleted or shortened

Response 4:

We deleted the sentences.
